# Molecular Classification and Interpretation of Amyotrophic Lateral Sclerosis Using Deep Convolution Neural Networks and Shapley Values

**DOI:** 10.3390/genes12111754

**Published:** 2021-10-30

**Authors:** Abdul Karim, Zheng Su, Phillip K. West, Matthew Keon, Jannah Shamsani, Samuel Brennan, Ted Wong, Ognjen Milicevic, Guus Teunisse, Hima Nikafshan Rad, Abdul Sattar

**Affiliations:** 1GenieUs Genomics, 19a Boundary St, Darlinghurst, NSW 2010, Australia; Abdul@genieus.co (A.K.); john@genieus.co (Z.S.); phillip@genieus.co (P.K.W.); jannah@genieus.co (J.S.); sam@genieus.co (S.B.); ted@genieus.co (T.W.); oggy@genieus.co (O.M.); guus@genieus.co (G.T.); 2School of Biotechnology and Biomolecular Sciences, Faculty of Science, The University of New South Wales, Sydney, NSW 2033, Australia; 3The New York Genome Center, 101 Avenue of the Americas, New York, NY 10013, USA; cgnd_help@nygenome.org; 4Institute of Integrated and Intelligent Systems, Griffith University, Nathan, QLD 4111, Australia; hima.nikafshanrad@griffithuni.edu.au (H.N.R.); a.sattar@griffith.edu.au (A.S.)

**Keywords:** machine learning, ALS, classification, interpretation, target identification

## Abstract

Amyotrophic lateral sclerosis (ALS) is a prototypical neurodegenerative disease characterized by progressive degeneration of motor neurons to severely effect the functionality to control voluntary muscle movement. Most of the non-additive genetic aberrations responsible for ALS make its molecular classification very challenging along with limited sample size, curse of dimensionality, class imbalance and noise in the data. Deep learning methods have been successful in many other related areas but have low minority class accuracy and suffer from the lack of explainability when used directly with RNA expression features for ALS molecular classification. In this paper, we propose a deep-learning-based molecular ALS classification and interpretation framework. Our framework is based on training a convolution neural network (CNN) on images obtained from converting RNA expression values into pixels based on DeepInsight similarity technique. Then, we employed Shapley additive explanations (SHAP) to extract pixels with higher relevance to ALS classifications. These pixels were mapped back to the genes which made them up. This enabled us to classify ALS samples with high accuracy for a minority class along with identifying genes that might be playing an important role in ALS molecular classifications. Taken together with RNA expression images classified with CNN, our preliminary analysis of the genes identified by SHAP interpretation demonstrate the value of utilizing Machine Learning to perform molecular classification of ALS and uncover disease-associated genes.

## 1. Introduction

Amyotrophic lateral sclerosis (ALS) refers to a group of rare neurological disorders in which nerve cells (neuron) functionality to control voluntary muscle movement such as chewing, walking and talking is jeopardized [1,2,3]. The disease results in a progressive loss of muscle strength leading to paralysis and eventually death [2]. Genetic aberration is one of the primary causes of ALS for many patients [2,4]. Most of these genetic aberrations are non-additive because of their interaction with each other which makes it challenging to be detected using classical available genotype-phenotype association approaches [2]. ALS is now recognized as a multidimensional spectrum disorder. Recently, deep learning techniques have been proven to be widely used for predicting genotype-phenotype associations and molecular ALS classifications [5,6,7,8]. The ability of deep learning models to effectively extract nonlinear relationships from a large number of samples for complex disorders has been reported in the literature [9,10].

The molecular ALS classification is also a very complex problem [11] and ideally would require thousands of samples [12] to train any deep learning algorithm. ALS is a rare disease, and it is challenging to find a large number of samples for research purpose [12,13]. Another factor that limits the availability of samples is the accessibility of affected tissues. ALS is a disease of motor neurons, which reside in the spinal cord and the brain [14]. Postmortem spinal cord, brain and cerebrospinal fluid are ideal tissue sources which either directly reflect the pathology of the disease or have interaction with central nervous system, but they are usually more difficult to access compared to tissue such as blood [15]. In a typical dataset of rare disease (ALS) samples, the number of samples is far less than the number of expressed genes [16,17], thus introducing the curse of dimensionality [18].

Another challenge with study of RNA expression, which is the gene transcription activity represented by the count of reads mapped to the gene in next generation sequencing data, is tissue heterogeneity and cell composition heterogeneity. Different human tissues have distinct RNA expression patterns [19]; furthermore, it has been found that ALS patients and healthy individuals have different cell compositions in same tissue [20], and these can be confounding factors in disease-associated gene expression identification. As we are using postmortem samples in the study, RNA quality can be easily impacted by sample collection time and storage condition, which in turn influences data quality and gene expression quantification. This is another confounding factor that makes the data analysis challenging.

Besides limited sample size, the curse of dimensionality and noise of the data, rare disorders data also suffer from a severe class imbalance problem [21,22]. In machine learning, one of the important criteria for higher classification accuracy is a balanced dataset [22]. Datasets with a large ratio between minority and majority classes face hindrance in learning using any classifier [22]. In order to cater for dimensionality curse and class imbalance for molecular classification of ALS, we propose an end-to-end machine-learning-based pipeline for ALS and control samples classification and interpretation. We used RNA expression data for control (60) and ALS (490) samples, and each sample is represented in RNA expression values. The RNA expression values for each sample are mapped to form a pixel value of an image, thus creating an image dataset. We utilized DeepInsight package [23] for image creation and a convolutional neural network (CNN) for classification. We used various subsets of RNA expression data with our classification model. Using this approach, even with a small size, highly noisy and severe class imbalance dataset, we achieved promising classification results. Our method achieves better performance for a minor class in comparison to other classical methods such as a fully connected neural network trained, random forest and support vector machines trained on RNA expression values directly.

In addition to molecular classification of ALS, we also employed SHAP (Shapley additive explanations) to interpret the prediction results of our classification module [24]. We identified the top 10 pixels with the highest SHAP values and investigated the genes which these pixels represented. The model found known ALS-associated genes and predicted potential new disease genes. we demonstrated the value of utilizing machine learning to perform molecular classification of ALS. In this study, we show that our image-based neural network approach is able to perform effective feature selection, learn nonlinear relationship in highly noisy data and identify biologically relevant molecular signals.

## 2. Materials and Methods

In this section, we first describe the dataset and the performance evaluation criteria used in this study. Then, we provide details about the developed pipeline and its major parts such as image creation module, classification module and post hoc interpretation module.

### 2.1. Datasets

New York Genome Center (NYGC) RNASeq data were used for this study, and RNA extraction, library preparation and sequencing were performed by NYGC under their protocol. In brief, total RNA was extracted from flash frozen postmortem tissues of ALS and control samples, using trizol/chloroform method, followed by Qiagen RNeasy minikit column purification. RNA was quantified using Nanodrop 2000 and Qubit™ 2.0 Fluorometer, and its quality was measured by RNA integrity number (RIN) scores on Agilent Bioanalyzer. Libraries were prepared using KAPA Stranded RNA-Seq Kit, then loaded to Illumina HiSeq 2500 sequencer for 2 × 125 bp paired-end sequencing.

After receiving the raw sequencing data in fastq format, we performed quality control using FASTQC [25], with mean quality value across each base position in the read and per-sequence quality scores as the main metrics for data quality evaluation. The sequences were pseudo aligned to reference genome of GRCh38 from Ensembl release 95 [26] by Kallisto [27] for RNA expression quantification. GTF file from Ensembl release 86 was used for gene region annotation. The transcript abundance quantified by Kallisto was used for downstream machine learning.

As sex chromosomes have different ploidies in different genders, we only use genes in autosomes, i.e., chromosomes 1–22 in our pipeline. We also excluded loci with multiple haplotypes in GRCh38 (such as MHC locus), where accurate expression quantification is challenging. As genes with low read count carry little information and can be caused by mapping errors, we conducted a case study which only used high-expression genes, which are genes with ≥10 reads in ≥10 samples, for model training. In another case study, we used only protein-coding genes for training, which are defined as protein-coding genes in Gencode Release 26 [28].

### 2.2. Evaluation Criteria

In order to measure the ALS classification performance of our developed pipeline, we used the following metrics: area under curve of receiver operating curve (AUC-ROC), specificity (SPE), sensitivity (SEN), negative predictive value (NPV), positive predictive value (PPV), accuracy (ACC) and Matthew’s correlation coefficient (MCC). It should be noted in this paper that negative class refers to control, and positive class refers to ALS. The details of these metrics are as follows:Area under curve of receiver operating curve (AUC-ROC): AUC-ROC takes into account all the threshold. The higher the value of AUC-ROC, the better the model is at distinguishing between classes. It can be computed by taking area under the curve for true-positive rate (TPR) on the y-axis and false-positive rate (FPR) on the x-axis for a given dataset. TPR which is also called sensitivity (SEN) describes how good the model is at classifying a sample as ALS when the actual outcome is also ALS. FPR describes how often a ALS class is predicted when the actual outcome is control.
(1)$$SEN=TPR=\frac{TP}{TP+FN}$$
(2)$$FPR=\frac{FP}{FP+TN}$$
where TP = True Positives, TN = True Negatives, FP = False Positives, FN = False Negatives, and SEN = Sensitivity.Specificity (SPE): SPE is the total number of true negatives divided by the sum of the number of true negatives and false positives. Specificity would describe what proportion of the negative class were correctly classified by our model.
(3)$$SPE=\frac{TN}{TN+FP}$$Negative predictive value (NPV): NPV describes the probability of a sample predicted as negative class to be actually as negative class.
(4)$$NPV=\frac{TN}{TN+FN}$$Positive predictive value (PPV): PPV describes the probability of a sample predicted as positive class to be actually as positive class.
(5)$$PPV=\frac{TP}{TP+FP}$$Accuracy (ACC): ACC is the fraction of prediction our model was correct about, i.e., it predicted positive class and negative class correctly.
(6)$$ACC=\frac{TP+TN}{TP+TN+FP+FN}$$Matthews correlation coefficient (MCC): MCC has a range from −1 to 1 where −1 indicates a completely wrong binary classifier while 1 indicates a completely correct binary classifier.
(7)$$MCC=\frac{TP*TN-FP*FN}{\sqrt{\left(TP+FP)(TP+FN)(TN+FP)(TN+FN\right)}}$$

### 2.3. Image Creation Module

We transform RNA expression data into pixels using the DeepInsight similarity technique and then trained a convolution neural network (CNN).

In Figure 1, DeepInsight for RNA expression data is used to create images of ALS and a control group sample. DeepInsight image creation process is shown in Figure 2. The RNA expression values were mapped to a 2D matrix such that the features which are similar to each other occupy nearby positions in the matrix. As shown in Figure 2a, a transformation T was applied on the genes feature vectors for each sample which creates a 2D matrix M. Features g1, g3, g6 and gd are closer to each other, which brings them into each other’s vicinity after applying the transformation T. On the other hand, gene feature g7 is different than the others and mapped to a very different location in 2D matrix. Figure 2b shows each step of the transformation T.

The first step is to find the location of each gene feature. For that purpose, similarity measuring technique or dimensionality reduction technique like t-SNE or kernel principal component analysis (kPCA) is applied sample-wise on the gene features data. This results in feature locations in 2D plane. Once the location of the gene features of samples is determined, then convex hull algorithm is used to find the smallest rectangle that contains all the points. Rotation is performed to obtain images in horizontal or vertical orientation only. Then the gene features are mapped to their respective positions obtained in the previous step. Thus a new image data is created for each sample where each pixel correspond to one or more gene features. In case of multiple genes very similar to each, their values are averaged out to obtain a pixel.

In specific implementation for RNA expression data as gene features in our case, we first normalized the expression values to a range of [0, 1]. We used normalization method named as norm-2 from DeepInsight [23]. In this normalization, the topology of features was reserved to some extent of DeepInsight [23]. After normalization, we used a python package of DeepInsight [29] with t-SNE method to generate single-channel images with a dimension of 380 × 380 in our final model. For the demonstration purposes, we show an image of 120 × 120 in Figure 1. Major portion of both ALS and control was similar, yet subtle changes can be observed, one of which is highlighted in Figure 1.

### 2.4. Classification Module

After obtaining image data from RNA expression data, we used convolutional neural network (CNN) to classify images into ALS and control. A convolutional neural network is a special type of neural network for the image data. CNNs can extract low-level features from images and compute more complex features as we go deeper in the networks [30,31]. Variants of CNN such as Inception, Alexnet and Resnet have been developed and employed as highly accurate image classification models [32]. In our particular case, as shown in Figure 3, there was an input, 2 conv blocks, then fully connected block followed by an output block. Input contained image data of 490 ALS and 60 control samples. Each sample was 380 × 380 single-channel image. There were two conv blocks concatenated after the input. Each of the conv block consisted of one 2D convolution layer followed by ReLU activation, max pooling and drop out as shown in Figure 3. The depth d of 2D convolution was 32 in first conv block and 64 in the second conv block. In fully connected block, a dense layer with 256 units along with ReLU activation was used after flattening the output of the conv block. A single dense unit followed by sigmoid function was applied at the end of the fully connected block. At the output, the sample was considered as ALS for prediction probability greater than 0.5 and control for prediction probability less than 0.5.

#### CNN Training

We used Keras deep learning framework [33] for developing and training the CNN model. We trained it for 500 epochs with an early stopping criteria. During the training, we used class weights for computing the loss function to cater for class imbalance in our data. ALS class was assigned with a weight of 0.56, and control was assigned with a weight of 4.57 using sklearn class_weight function [34]. We used ADAM optimizer [35] with a binary cross entropy loss function from Keras [33] for training our CNN model.

### 2.5. Classical Machine Learning Methods

We used random forest (RF), support vector machines (SVM) and a fully connected neural network (FCNN) with 33,153 RNA expression features related to autosomal genes directly to compare their results with our method.

#### 2.5.1. Random Forest (RF)

Random forests are a combination of tree predictors, such that each tree depends on the values of a random vector sampled independently and with the same distribution for all trees in the forest [36]. We used RF from scikit-learn machine learning library with its default parameters [37].

#### 2.5.2. Support Vector Machines (SVM)

Support vector machines (SVM) belongs to a class of supervised machine learning methods. It attempts to find a line/hyper-plane (in multidimensional space) that separates classes of data under observation or ranges for regression [38,39]. We used SVM from scikit-learn machine learning library with its default parameters [37,40].

#### 2.5.3. Fully Connected Neural Networks (FCNN)

We also compared our method with a fully connected neural network (FCNN). FCNN can be viewed as a complex mapping function, where the fundamental unit of a FCNN is called a neuron. It takes input and computes the output after applying non-linearity and gradient descent based back-propagation algorithm [41]. In specific implementation for this study, FCNN consisted of two fully connected layers with 200 neurons in each, a second last layer with 10 neurons and an output layer with one neuron. We placed a dropout rate of 0.5 after each hidden layer. We used Keras deep learning framework [33] for training. We trained it for 100 epochs with an early stopping criteria. We used ADAM optimizer [35] with a binary cross entropy loss function from Keras [33] and a batch size of 32 with a learning rate of 0.001.

### 2.6. Post Hoc Interpretation Module

Deep learning methods such as CNN are black-box in nature and extremely difficult to interpret [42,43]. These methods are capable to answer the “what” question about certain prediction but fails to give an answer to the “why” question [43,44]. Understanding why a model makes a certain prediction can be as crucial as the prediction’s accuracy in many applications [24]. In this study, we used SHAP (Shapley additive explanations) to interpret the prediction results of our classification module. SHAP assigns each feature an importance value for a particular prediction. It connects game theory with local explanations and uniting several previous methods [44,45,46,47]. SHAP represents the only possible consistent and locally accurate additive feature attribution method based on expectations [24].

In this study, once we have trained and tested our model, we employed Deep SHAP package [48] to interpret the prediction outcome of our classification module. In the first step, we selected a distribution of background 200 random samples out of the input image data to take expectations over. Then, we used the selected background distribution along with the trained model to obtain SHAP values for each pixel in a sample as shown in Figure 4. Red pixels represent positive SHAP values that increase the probability of the class, while blue pixels represent negative SHAP values that reduce the probability of the class. The sum of the SHAP values equals the difference between the expected model output (averaged over the background dataset) and the current model output. After obtaining SHAP values of each pixel, we selected the top ten pixels with the highest SHAP values and mapped them back to specific genes forming those pixels.

## 3. Results and Discussion

In this section, we present the classification performance of our method with different image resolutions and RNA expression features related to various sets of genes. We also show our method’s performance compared with classical machine learning models and the top 10 gene extracted using SHAP interpretation of our method. Finally, we investigate the functional and disease association of the extracted genes and then discuss some potentially identified new genes.

### 3.1. Samples and Quality Controls

Fastq files of 550 samples (490 ALSs and 60 controls) were downloaded from NYGC database; all of them passed the default quality control criteria of mean quality value in the read and per sequence quality scores of FASTQC; and all samples were used for downstream analysis. The samples are from 114 ALS patients and 19 control individuals. The majority of patients and control individuals donate multiple tissues, including cerebellum (*n* = 98), frontal cortex (96), motor cortex (118), occipital cortex (58), and spinal cord (163) (Appendix A), and the composition of tissues is similar between ALSs and controls (Appendix A). We quantified the expression of 39,429 genes (33,153 autosomal and 6276 non-autosomal genes), and the expression data of autosomal genes were used for model training.

RIN is one of most commonly used metrics for RNA quality control. It has been shown that RIN values impact RNA sequencing data quality and gene expression quantification [49]. The suggested threshold of RIN value for sample inclusion varied in different studies, and it can be as high as 8 [50] and as low as 3.95 [51]. The samples in this study have RIN values ranging from 2.2 to 9.9 (Appendix A), and we did not filter out samples with low RNA quality using a RIN threshold, as it can reduce the power of the analysis [49]; by contrast, we just used all samples for model training and left the task of distinguishing biological signals from RNA quality confounding effects to the model. It should be noted that even though we did not apply any filter on the RIN values, there should not be any bias in RIN values between two groups; otherwise, it can confound the prediction. There is no significant difference in the distribution of RIN values in two groups (*p*-value = 0.70, *t*-test) as shown in Appendix A.

### 3.2. Classification Performance for Various Image Resolutions

We present the effects of different image resolutions on the classification performance of our method. For this purpose, we used RNA expression features of 33,153 autosomal genes. We performed 12-fold cross-validation experiments with image resolutions starting from 50 × 50 till 380 × 380 as shown in Figure 5. AUC and ACC improves with higher image resolutions. The highest value of AUC: 0.927 is obtained for the resolution of 320 × 320, whereas for that of ACC: 0.827 is obtained for the resolution of 380 × 380. For F1 and MCC, we observe an initial improvement with a dip in between around 220 and 250 followed by an improvement until 380 × 380. The highest values for both F1: 0.813 and MCC: 0.639 are obtained for the resolution of 380 × 380. As the data are highly imbalanced with ALS as major class and control as minor class, we see higher sensitivity with little variations as compared to specificity for all image resolutions. SPE increases continuously with higher image resolution with a highest value of 0.706 for 380 × 380. The ability of our model to correctly identify minor class which is control in our case improves significantly with higher resolution images. For PPV on the other hand, there is a slight improvement with a highest value of 0.963 for 380 × 380. The highest value for NPV however occurs at a resolution of 180 × 180, which is 0.790. Moreover, with higher resolutions, the standard deviation for 12 folds increases, as shown by the black error bars in Figure 5.

### 3.3. Comparison with Classical Models

In order to investigate the effectiveness of our method, we compared its results with classical machine learning methods such as random forest, support vector machine and fully connected neural networks. As shown in Table 1, our method performs significantly better in most of the classification metrics. Specifically in classifying the control class correctly, our method proves to be very robust as shown by SPE value in Table 1. Both RF and SVM achieves very poor performance for classifying the minor class correctly. FCNN performs relatively well though as compared to RF and SVM for SPE. A relatively higher value of AUC for RF and SVM is strongly influenced by the major class, which is most of the time predicted correctly. AUC for our method is the highest as compared other classical methods. Nearly all the methods are performing well for classifying the major class as shown by the SEN value in Table 1. SVM achieves the highest NPV followed by RF, FCNN and our method, respectively. For PPV, ACC, MCC and F1, our methods achieves 3.54%, 20.37%, 33.68% and 12.44% improvement over the second-best FCNN method.

### 3.4. Case Study of Classification Performance with High Count and Protein-Coding Genes

In the initial phase of this study, we used expression values of all available 33,153 autosomal genes to train our pipeline with 380 × 380 image resolutions. We observed that even though higher resolution of images give better classification performance, it also increases the computational complexity and run-time of the pipeline. Therefore, we chose the resolution of 350 × 350 for two case studies to investigate the effectiveness of high-expression genes and protein-coding genes.

High-expression genes: Many of the 33,153 autosomal genes only express in very few samples and carry little information; thus, we filtered out those genes with low expression and only included genes with a high read count in a certain number of samples. For that purpose, we used a threshold of 10, i.e., at least 10 samples across our training data have a read count of 10 or higher. Through this filtering strategy, we obtained a total of 18,194 high expression genes. The RNA expression values of these high read count genes were converted into 350 × 350 images and subsequently used in CNN training for classification.Protein-coding genes: As including non-protein-coding genes in the training data may only increase the model complexity but bring little benefit to the model, thus in the second case study, we also selected RNA expression data of 19,724 protein-coding genes, converted them into images with resolution of 350 × 350 and evaluated the performance of CNN model trained with those images.

As shown in Table 2, for all the metrics, RNA features for high-expression genes substantially improve the classification performance as compared to 33,153 autosomal gene expression results shown in Table 1. Protein-coding genes however show a slight decrease in the classification performance.

### 3.5. SHAP Interpretation

We used SHAP interpretation [24] to investigate the role of each gene in classifying ALS samples using our developed model. It should be noted that each pixel of the image may contain one or more gene expression values. It should be noted that for SHAP interpretation, we used the model obtained with high-count gene RNA expression features with image resolution of 350 × 350. For each prediction of our model discussed previously, we identified top 10 pixels having highest SHAP values. We identified 12 genes (Appendix A) which appeared in the top 10 pixels in more than 200 samples. We are currently investigating 10 of the genes further; however, two genes in our dataset, Survival of motor neuron-1 (SMN1) and SMN2, have been previously classified as associated with disease in ALS [52]. SMN1 and SMN2 are paralogous genes as the result of an inverted duplication [53]. The SMN protein has a myriad of roles in motor neuron function and is critical for the regulation of transcription and RNA maturation, axonal trafficking of RNA transcripts, facilitating the binding of RNA-binding proteins to mRNA transcripts and regulating cytoskeletal dynamics [54]. In addition, SMN is involved in protein degradation pathways by its actions in autophagy and the ubiquitin-proteasome system (UPS) and may contribute to mitochondrial function through regulation of splicing, translation or mRNA transport of genes crucial for mitochondrial homeostasis [54]. Therefore, the SMN protein encoded by SMN1 and SMN2 contributes to many distinct pathways implicated in the pathology of ALS.

In addition, SMN interacts with, and its properties are modulated by, other proteins known to contribute to ALS pathogenesis, such as FUS, SOD1 and TDP-43 [55]. The interaction between ALS and FUS is enhanced by ALS-associated mutations in FUS, causing these proteins to form a stable complex [56]. As a result, SMN is mislocalized and sequestered into cytoplasmic FUS aggregates, leading to a decrease in SMN in the axons of neurons and subsequent axonal defects [56]. These aberrations also result in the loss of small nuclear bodies, dysregulated small nuclear ribonucleoprotein (snRNP) assembly and defects in downstream RNA processing [57,58]. Similarly, SOD1 variants cause the mislocalization of SMN and disrupt the formation of nuclear bodies [59,60]. In vitro over-expression of SMN enhances chaperone activity and protects cells from mutant SOD1 toxicity [61]. Furthermore, over-expression of SMN in SOD1 or TDP-43 mutant mice ameliorates disease [62,63], while SOD1 mutant mice exhibit accelerated disease severity when SMN is depleted [64]. Therefore, the currently available evidence indicates that, in addition to its function in relevant biological pathways affected during ALS, the SMN protein may cooperate with other ALS-associated genes to coordinate and modify the disease phenotype of ALS. It should be noted that there are changes in cell-type composition in ALS as found in previous studies [20,65]; therefore, in the interpretation of top genes found in our model, we should be cautious that they can arise from this confounding factor. Taken together, our preliminary analysis of the genes identified by SHAP interpretation demonstrate the value of utilizing Machine Learning to perform molecular classification of ALS and uncover potential disease-associated genes.

## 4. Conclusions

In conclusion, we developed a deep learning framework, which took full advantage of image recognition ability of convolutional neural network by transforming the gene expression data of ALS into images and then used them for neural network training. We showed that the model effectively extracted disease associated features and learned the nonlinear gene-disease relationship in highly noisy, heterogeneous and imbalanced data, and we also showed its superior performance in disease clarification over other machine learning algorithms. We interpreted the model with SHAP and successfully identified known disease associated genes, and some potential new disease genes, which demonstrated the potential of our model in the new biomarker and drug target identification in complex disease research.

## Figures and Tables

**Figure 1 genes-12-01754-f001:**
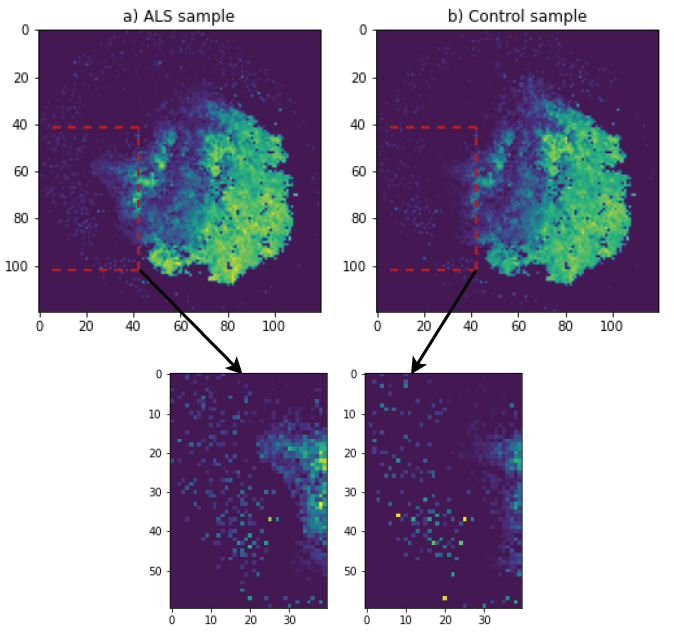
ALS and control sample images with 120 × 120 resolution obtained using DeepInsight for demonstration purposes.

**Figure 2 genes-12-01754-f002:**
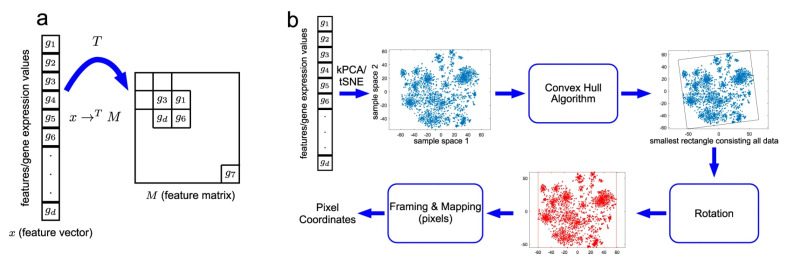
DeepInsight pipeline. (**a**) An illustration of transformation from feature vector to feature matrix. (**b**) An illustration of the DeepInsight methodology to transform a feature vector to image pixels. Image taken from DeepInsight [23].

**Figure 3 genes-12-01754-f003:**
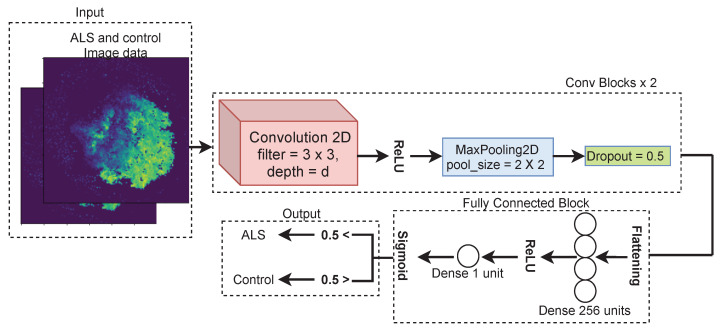
CNN architecture for classifying ALS and control images.

**Figure 4 genes-12-01754-f004:**
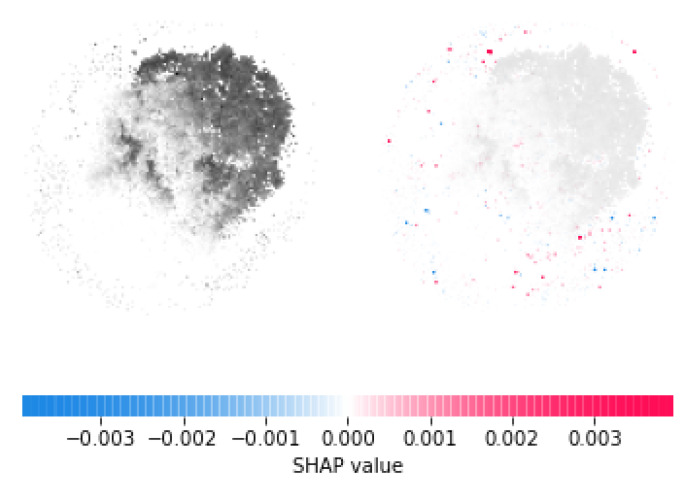
Left side is gray-scale image of an ALS sample. Right side shows highlighted pixels in the image with SHAP values.

**Figure 5 genes-12-01754-f005:**
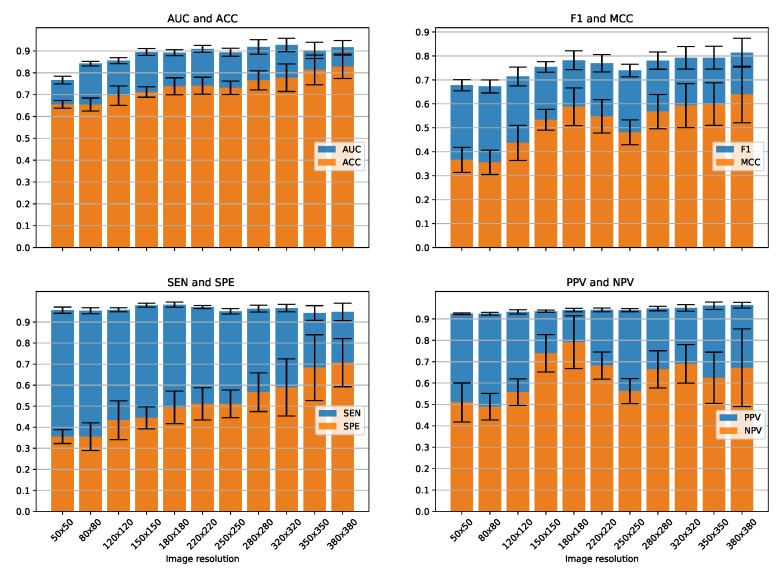
12-fold cross-validation performance for creating images of various resolutions.

**Table 1 genes-12-01754-t001:** 12-fold cross-validation performance comparison with classical machine learning methods such as random forest (RF), support vector machines (SVM) and fully connected neural network (FCNN). For our method while comparing with classical methods, we used images with resolution of 380 × 380 and all 33,153 RNA expression features of autosomal genes.

Method	AUC	SPE	SEN	NPV	PPV	ACC	MCC	F1
**Our method**	**0.917** ± 0.03	**0.707** ± 0.11	0.947 ± 0.04	0.671 ± 0.18	**0.963** ± 0.01	**0.827** ± 0.05	**0.639** ± 0.11	**0.813** ± 0.06
RF	0.831 ± 0.04	0.155 ± 0.05	0.994 ± 0.00	0.798 ± 0.19	0.906 ± 0.00	0.575 ± 0.02	0.319 ± 0.09	0.602 ± 0.04
SVM	0.866 ± 0.05	0.083 ± 0.03	**1** ± 0	**1** ± 0	0.899 ± 0.00	0.541 ± 0.01	0.270 ± 0.04	0.549 ± 0.02
FCNN	0.805 ± 0.04	0.4 ± 0.12	0.974 ± 0.02	0.692 ± 0.20	0.930 ± 0.01	0.687 ± 0.06	0.478 ± 0.15	0.723 ± 0.07

**Table 2 genes-12-01754-t002:** 12-fold crossvalidation classification performance with a resolution of 350 × 350 high-expression and protein-coding genes RNA features.

RNA Features	AUC	SPE	SEN	NPV	PPV	ACC	MCC	F1
High count genes	**0.964** ± 0.04	**0.776** ± 0.12	**0.978** ± 0.00	**0.809** ± 0.00	**0.973** ± 0.01	**0.877** ± 0.06	**0.767** ± 0.10	**0.882** ± 0.05
Protein-coding genes	0.910 ± 0.04	0.646 ± 0.13	0.968 ± 0.02	0.720 ± 0.12	0.957 ± 0.01	0.807 ± 0.07	0.643 ± 0.13	0.819 ± 0.06

## Data Availability

The raw RNA sequencing data used in this study was download from The ALS Consortium of New York Genome Center (https://www.nygenome.org/als-consortium/). The gene expression read count data isn’t available due to data share policy in the data agreement with the data provider.

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
