# Peer review of "Molecular Classification and Interpretation of Amyotrophic Lateral Sclerosis Using Deep Convolution Neural Networks and Shapley Values"

_genes, 2021, doi:10.3390/genes12111754_

Round 1

Reviewer 1 Report

Karim et al. described a deep learning model to predict amyotrophic lateral sclerosis (ALS) and nominate important genetic features in the trained deep learning model using Shapley values. The authors reported the trained deep learning model from the DeepInsight approach outperformed other conventional machine learning models. However, there are a few suggestions that can be addressed to make the study clearer.

 (1) The authors reported the study contained RNA expression from 60 control and 490 ALS samples. Can the authors also elaborate on how many patients cover these samples? It is confusing to see the cohort size and whether a patient donates multiple samples? It is also confusing whether 60 control samples come from 60 people and whether these are matched adjacent tissues?

(2) The authors reported the model performance on 12-fold cross-validation. However, it is challenging to decide its actual clinical use without informing the model performance on the independent test data set. Can the authors split the dataset into 80% training data and 20% holdout data and apply 5-fold cross-validation on the 80% training data? In this approach, it will be fair to see the performance of the DeepInsight model on the holdout data that have not been used in training.

(3) KIF5A is a reported novel ALS gene (Nicolas et al., Neuron, 2018). However, this gene does not appear on the top 10 genes reported from Shapley values. Can authors also declare the rank of the KIF5A gene and its Shapley value in the result? Can authors also elaborate on the difference between their result and top ALS genes reported in Nicolas et al., Neuron, 2018? 

Author Response

Dear reviewer,

Thanks for your valuable comments. We have provided point by point response and revised our manuscript as per your comments. Please find the attached response letter.

Thanks

Reviewer 2 Report

The methodology described in this manuscript was well introduced and very interesting. In a very much non-standard approach, the authors apply a transformation of RNAseq based transcript count tables which allows them to then apply convolutional neural nets for a classification task. They benchmark their approach against direct analyses of transcript count data using other popular machine learning algorithms (including an alternative neural net). They report that their CNN approach achieves better performance in predicting whether post mortem brain RNAseq samples originated from ALS patients or non-neurological disease controls. They then apply deep SHAP to aid interpretation of their model by scoring genes based on the contributions they make to model prediction. To me the implications of the results for the field of ALS are either oversold or could at least benefit from some clarifications (outlined in comments below). However I think the comprehensiveness of the methodological evaluation holds in either case and would be of interests to readers. I would propose only minor clarifications of the ALS claims is likely to be needed.

Main comments
1) The molecular classification task being performed here does not appear very relevant to the real world. After reading "Most of the non additive genetic aberrations responsible for ALS make its molecular classification very challenging", I thought there would be an attempt to distinguish different ALS sub types or ALS vs ALS mimic disorders, but the focus appears to be comparing ALS brain tissue to non-neurological disease controls. If the authors would like to pitch that their model has a real world use-case it would help to clarify what exactly that is (eg rather than routine use of the model to distinguish ALS patients vs controls, the objective is more biomarker discovery?)
2) ALS is a neurodegenerative disease and by definition the cell type composition is different in patients vs controls. It would be expected that these differences dominate any case-control comparison, in which case the claim that the model identifies disease relevant genes seems too strong (ie it seems a stretch to consider every single cell marker that might differ in patients vs controls as a "disease associated gene" or "drug target"). A short note to clarify the limitations of the model and how it can be interpreted would be useful, and standard.
3) The dataset includes many tissues, presumably many of these tissue profiles originate from the same individual. In which case the authors should discuss a little that samples are not independent and elaborate whether there were any case-control biases in tissue representation.
4) The authors report that they "left the task of distinguishing biological signals from RNA quality confounding effects to the model". It seems important to clarify if there was any bias in RIN value in cases vs controls. Otherwise, why would we assume that the model had not learned to predict case control status simply based on RNA quality?

Minor notes
1) Figure 1 is taken from another paper. Presumably with permission
2) I was not sure about the statement that "most genetic abberations are non-additive"? Is this meant to infer that clear examples of statistical interaction / non-linear effects have been demonstrated in ALS? While this may well be the case I am not aware that it has been convincingly demonstrated

Author Response

(The authors gave the same response as above.)
